# Improving Coherence and Consistency in Neural Sequence Models with Dual-System, Neuro-Symbolic Reasoning

**Maxwell Nye**[*]
MIT

**Michael Henry Tessler**
MIT
DeepMind

**Joshua B. Tenenbaum**
MIT

**Brenden M. Lake**
NYU
Facebook AI Research

## Abstract

Human reasoning can be understood as an interplay between two systems: the intuitive and associative ("System 1") and the deliberative and logical ("System 2"). Neural sequence models—which have been increasingly successful at performing complex, structured tasks—exhibit the advantages and failure modes of System 1: they are fast and learn patterns from data, but are often inconsistent and incoherent. In this work, we seek a lightweight, training-free means of improving existing System 1-like sequence models by adding System 2-inspired logical reasoning. We explore several variations on this theme in which candidate generations from a neural sequence model are examined for logical consistency by a symbolic reasoning module, which can either accept or reject the generations. Our approach uses neural inference to mediate between the neural System 1 and the logical System 2. Results in robust story generation and grounded instruction-following show that this approach can increase the coherence and accuracy of neurally-based generations.

## 1 Introduction

Despite recent success, neural sequence models often fail to produce consistent and coherent generations. When generating stories, language models may forget the attributes of specific characters (such as personality and background information) (Welleck et al., 2018), ignore previously established relationships between characters (such as family relationships) (Sinha et al., 2019), or otherwise contradict prior statements (Brown et al., 2020). Similarly, neural models can make statements that contradict basic world knowledge or the logical entailment structure of known facts.

Lake & Murphy (2020) illustrated several of these issues with GPT-2 (Radford et al., 2019). When given prompts of the form *"A dolphin is a ___"*, GPT-2 predicts that the most likely answer is "mammal", "fish", or "bird" depending on small differences in the wording of the prompt. In another example, GPT-2 states that unicorns have "four horns," directly after implying that unicorns only have one horn. Upon diagnosing such issues, it is unclear how to apply a targeted fix to the model, especially if retraining or fine-tuning is impractical.

In this work, we draw on insights from cognitive science, especially from "dual process" theories of reasoning (Evans, 2003), to explore how neural sequence models can better interface with prior knowledge and be made more coherent and consistent. According to dual process theories, human cognition can be understood as an interplay between a more intuitive and associative "System 1" and a more deliberative and logical "System 2." Within this broad framework, automatic actions are driven by System 1, whereas System 2 engages for more deliberative control: for example, judging the validity of a logical argument that requires multiple steps of reasoning (Kahneman, 2013).

---

[*]Correspondence to `mnye@mit.edu`. Work primarily done during an internship at Facebook AI Research.

35th Conference on Neural Information Processing Systems (NeurIPS 2021).

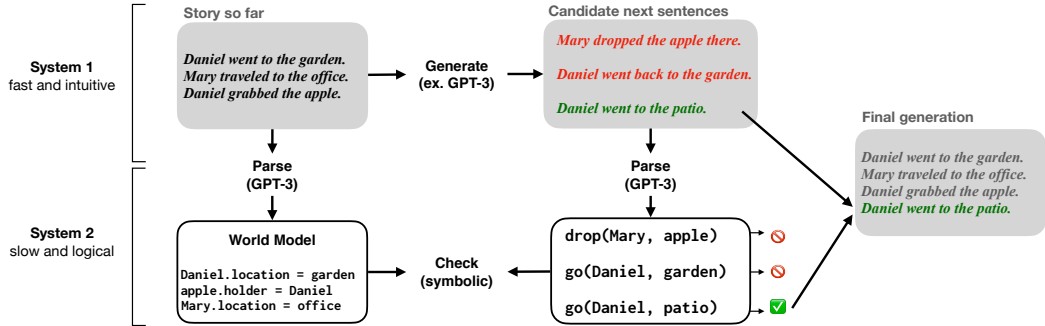

Figure 1: Schematic of dual-system approach to text generation. Conditioned on previous text, a "System 1" neural generation model produces candidate next sentences. Semantic parses for each candidate are generated via few-shot parsing from GPT-3 and compared to a minimal world model to check consistency. Only candidates consistent with the world model state are incorporated into the final generation.

The prominent neural language models of today are single systems, with weaknesses akin to those exhibited by the human System 1. For example, the cognitive reflection test (CRT) (Frederick, 2005) is a classic probe of System 1 vs. System 2 reasoning in humans. Participants answer a set of simple questions that have superficially compelling, but logically invalid, answers. These incorrect answers are often generated as a first "gut" response (putatively, by System 1 intuitive thinking); upon reflection, however, participants often realize that their responses were not logically or mathematically consistent (via more explicit System 2 reasoning). Consider the CRT problem on the left below:

| | Total cost in prompt | GPT-3 response |
|---|---|---|
| *A ball and a bat cost **$1.10**.* | $1.10 | 10 cents |
| *The bat costs one dollar more than the ball.* | $1.20 | 20 cents |
| *How much does the ball cost?* | $1.30 | $0.30 |
| | $1.70 | $0.70 |

Reading quickly, you might be tempted to say the ball costs 10 cents. Most participants give this response, in fact, especially if they are under time pressure or have limited attention (Kahneman, 2013). Of course, if the bat is $1.00 more than the ball, and the ball costs 10 cents, then the *total* cost would be $1.20. The correct answer is that the ball costs 5 cents. Notably, in this and other classic CRT problems, GPT-3 (Brown et al., 2020) predicts the same "gut" response (prediction in red above; the table above shows that adjusting the price in the prompt also leads to similar effects; see Appendix Figure 8 for more CRT examples). GPT-3 appears vulnerable to the same sort of intuitive, unsystematic pattern recognition errors as humans—in this case, incorrectly subtracting one dollar from $1.10, without confirming that the answer satisfies each of the problem constraints.

Numerous studies have shown that engagement of System 2-style effort can help "override or inhibit default responses emanating from System 1" (Evans, 2003), correcting inconsistent or un-systematic intuitive impulses. For example, when System 2 is engaged by asking people to take more time to respond, people's accuracy improves on the CRT task above (Kahneman, 2013). It has been argued that integrating System 2 processing could similarly improve AI systems (Goyal & Bengio, 2020; Garcez & Lamb, 2020), and here we explore this idea as applied to neural sequence models.

In this work, we take inspiration from dual process theories to explore a neuro-symbolic generation system, wherein predictions from a neural model are treated as System 1 proposals, and a logical, deliberative System 2 filters these proposals for consistency and soundness (see Figure 1). We further take inspiration from the fact that humans often do not need explicit supervision to reason about new problems or domains (e.g., see human evaluation task in Section 4.2) and require that the System 2 module not need additional problem-specific training, especially on example contradictions or commonsense violations. People can handle novelty by reconfiguring, rather than retraining, their internal models (Lake et al., 2017), and we strive to build machine systems capable of the same. We show how a lightweight, easy-to-implement System 2 model can help improve coherence and consistency by adding a small amount of symbolic reasoning.

We tackle two kinds of domains: text generation and instruction following. In both cases, we construct generative models over sequences by using a neural generation model to propose candidate generations and a symbolic world model that can accept or reject the generations and resample proposals if necessary. We first illustrate the approach by generating short stories based on the bAbI dataset (Weston et al., 2015); this pedagogical, synthetic example illustrates how basic commonsense knowledge of objects, agents, and places can inform a text generation model. We then test our

approach on rich, natural language vignettes based on CLUTRR (Sinha et al., 2019), focusing on ensuring consistency of family and interpersonal relationships. In both text generation domains, we interface between the explicit logical knowledge/reasoning of System 2 and generations of System 1 using a few-shot learning approach with state-of-the-art neural language models (GPT-3), which requires no additional training or fine-tuning. Even using off-the-shelf transformers and symbolic solvers, our dual-system model improves the consistency and coherence of text generations as measured by human judges. We test our approach also on instruction following, showing how goal-prediction models and execution models can easily be combined to achieve improved performance in low-data regimes. We show improvements over previous work in the gSCAN grounded compositional challenge (Ruis et al., 2020); a dual-system model requires much less data to train than previous models, and achieves higher accuracy and stronger generalization. Overall, our findings indicate that neuro-symbolic, dual process models are a promising means of addressing longstanding problems of robustness and consistency in neural sequence models.

## 2 Related Work

Our approach incorporates **semantic parsing** (Liang, 2016) as a component of a generative process, where neural generation is used in conjunction with parsing techniques. In our text generation experiments, we employ GPT-3 to perform few-shot semantic parsing without fine-tuning. Related work includes few or zero-shot semantic parsing using pre-training techniques and paraphrasing (Su & Yan, 2017; Herzig & Berant, 2020). It also includes semantic parsing systems trained either without supervision (Liang et al., 2017; Mou et al., 2017; Muhlgay et al., 2019), or with synthetic language data (Marzoev et al., 2020; Xu et al., 2020b).

One popular technique for improving neural generations is **generate-and-rerank**, wherein one model generates proposals and another reranks them. This broad approach has been used in image generation (Ramesh et al., 2021), text generation (Holtzman et al., 2018; Shen et al., 2019; Deng et al., 2020), dialogue systems (for control, coherence and safety (Welleck et al., 2018; Smith et al., 2020; Nie et al., 2020; Xu et al., 2020a)), and instruction following (Kurita & Cho, 2020). Reranking is generally used to improve outputs with respect to relatively broad, holistic criteria. Here, our goal is to make generation robust to particular types of logical errors by pruning with respect to explicit symbolic constraints. Our approach can thus be considered closely related to techniques which employ explicit search to find generations satisfying particular logical constraints. Similar methods, such as guess-and-check or beam search pruning, have had success in neural program synthesis (Devlin et al., 2017; Nye et al., 2020).

Recent work in NLP has used **template-based planning**, in which a model generates text by first generating a plan or skeleton, and filling in the missing words to produce naturalistic text (Xu et al., 2018; Hua & Wang, 2020). To generate stories, Martin et al. (2018) parses previous sentences into events and does planning in event space. Our work extends previous entity/relation/event planning in that the world model is not used for planning, but rather for post-checking candidate generations. Structured parsing of this type is also related to dialog tracking techniques such as slot-filling (Pieraccini et al., 1992). In our work, fully compositional logical facts are extracted from utterances. It is therefore more closely related to systems which extract programs from dialogue, such as Andreas et al. (2020).

Recent work has also studied incorporating symbolic constraints into a neural decoding strategy in the context of natural language. Miao et al. (2019) introduce an MCMC-based inference-time **propose-and-reject** strategy for satisfying constraints. They test on constraints such as paraphrase and grammatical error correction. Lu et al. (2020) introduces "NeuroLogic decoding," which uses logical constraints on neural language models to produce generations which contain (or do not contain) required (or forbidden) keywords. In these works, the constraints are *lexical* or based on word/sentence similarity (and provided in the problem setup for Lu et al. (2020)), whereas we study *logical* constraints on the world state decoded directly from observations or generations at test time. Other approaches for solving reasoning tasks end-to-end include Goyal et al. (2021), Serafini & d'Avila Garcez (2016), and Schlag & Schmidhuber (2018).

# 3 Integrating System 1 and System 2

We introduce our dual-system approach using examples from the bAbI domain (Weston et al., 2015), which we also use to perform diagnostic experiments. Consider generating a simple story involving people, places and objects, such as (from Figure 1):

> *Daniel went to the garden. Mary traveled to the office. Daniel grabbed the apple.*

A model tasked with generating such stories must juggle several simultaneous demands: staying on topic and maintaining consistency of style and other textural elements (for which people rely on System 1), as well as maintaining consistency with previous statements and commonsense knowledge (for which people rely on both systems). Consider continuing the story with one of the following:

> (a) *Daniel went to the patio.*       (b) *Mary dropped the apple there.*

Sentence (a) is reasonable; sentence (b) is not because it is Daniel, not Mary, who has the apple. During generation, how might a model distinguish between these candidates? Perhaps a well-trained neural language model could track constraints of these sorts. Neural language models to date, however, often violate these types of commonsense, hard constraints without a large high-quality corpus or explicit training on detecting violations of commonsense (Sinha et al., 2019).

We address this problem by decomposing text generation into two parts: candidate generation facilitated by deep neural networks and a logical pruning process implemented via a separate symbolic module. Consider again the example above. To ensure consistency, our model would extract from the text the features of the world that are subject to the hard, logical constraints, such as the location of objects and who is holding them. These constraints can then be checked against an explicit representation of current state of the world. For sentences (a) and (b), the system would extract and `go(Daniel, patio)` and `drop(Mary, apple)`, respectively. A **minimal world model** would track the state of the apple, such that it maintains `apple.holder = Daniel` (or equivalently, `Daniel.inventory = [apple]`). When such a model is given a parse of a candidate generation, `drop(Mary, apple)`, the mismatch between the current state and the proposed change would cause a violation, and the candidate generation will be rejected.

The main steps of our general approach are illustrated in Figure 1: generate proposals from a **System 1 proposal model**, extract facts with a **fact extraction model**, and filter proposed generations by ensuring that they satisfy the constraints given by the extracted facts and the **minimal world model**.

**System 1: Generation.** We use neural sequence models to produce System 1 generations. In text generation domains, we use a large, pre-trained model that can be fine-tuned or conditioned via a short prompt to generate relevant text. Text sampled from the System 1 model will be treated as candidate utterances, which will be parsed and filtered by System 2 (described below). For the bAbI examples, we use GPT-3 as our **System 1 proposal model** through few-shot prompting with 10 example bAbI stories as context, generating a new story one candidate sentence at a time.

**System 2: Fact extraction.** A fact extractor, or parser, is used to mediate between the System 1 candidate proposals and the minimal world model within System 2. In our text generation domains, we use a pre-trained GPT-3 model without fine-tuning to perform parsing.

For bAbI, our prompt consist of an initial descriptive sentence *"Please parse the following statements into commands. The available commands are pickup, drop, and go."* and a small set ($< 10$) of representative semantic parsing examples (input = sentences; output = correct parses, such as `go(Bob, roof)`). The parse of each utterance is produced via few-shot prompting (Brown et al., 2020): the utterance is added to the end of the prompt, and the subsequent GPT-3 generation is interpreted as the target parse. We found that this simple parsing technique works well and could easily be applied to other parsing-based tasks, as in Shin et al. (2021). The parsing prompts are reproduced in full in the Appendix. As discussed in Section 5, for the gSCAN instruction following domain, fact extraction is performed with a learned goal location prediction model.

**System 2: Minimal world model.** We use a lightweight, incomplete description of the state of the world as a world model in each domain, e.g., commonsense information about the people, objects and locations (Figure 1). The goal is not to track and verify all the possible information; instead, we aim for minimalism, capturing just a few commonsense (or application-critical) variables that we want to ensure are correct. The world model facilitates tracking of long-range logical dependencies and logical consequences, especially those which are not readily decodable from surface forms. The

world model also lets us integrate rule-based world-knowledge without retraining (and without the need for a large set of labeled examples).

For the bAbI examples, the minimal world model keeps track of the people, locations and objects introduced in the story so far (Figure 1). This encodes constraints on possible actions related to human core knowledge competencies (objects, agents, places) present early in human development (Spelke & Kinzler, 2007); specifically, a person or object can only be in one place at a time, an object can only be possessed by a single person at a time, a person cannot "go" to a room they are already in, and a person cannot pick up an object if it is in a different room. See the Appendix for details.

**Search.** At generation time, the interaction between System 1 generation and System 2 parsing yields a neuro-symbolic, guess-and-check search strategy. In a text generation scenario, where text is sampled from the model, our dual-system model improves upon a naive, neural-only sampling method by using the System 2 model to reject candidate utterances which are incompatible with the current state. When a candidate is rejected, a new candidate utterance is sampled from the System 1 model, which is again checked by System 2. This process repeats until a candidate utterance is accepted by System 2 (i.e., the utterance is compatible with the world state). This procedure allows the model to effectively search the space of candidate utterances, guided by the logical constraints from the minimal world model. In this work, we use straightforward probabilistic sampling to illustrate that the approach works with even a very simple search mechanism. We imagine that the search procedure could be further optimized by applying, for example, beam search or stochastic beam sampling.

**Diagnostic bAbI experiments.** We use Task #2 from bAbI as a diagnostic test for our neuro-symbolic dual-system model. As shown above, this task consists of synthetically-generated short stories involving people, places and objects, and questions concerning the locations of objects in these stories. We investigate performance on both question answering (QA) tasks and story generation. For the QA tasks, we parse each sentence in the story to encode each fact into the world model and parse the final question to query the world model, returning the answer given by the world

**GPT-3 only:**
*John went to the bedroom.*
*John picked up the apple there.*
*Mary took the apple there.*
*Mary travelled to the office.*
*Daniel went back to the garden.*
*Mary went to the bedroom.*
*John went to the bedroom.*
*Sandra went to the bedroom.*
*Sandra travelled to the office.*
*Mary went back to the office.*
*Where is the apple? A: office*

**GPT-3 + world model:**
*John went to the bedroom.*
*John picked up the apple there.*
*Mary travelled to the office.*
*Daniel went back to the garden.*
*Mary went to the bedroom.*
*Sandra went to the bedroom.*
*Sandra travelled to the office.*
*Mary went back to the office.*
*Where is the apple? A: bedroom*

Figure 2: Example bAbI stories generated by GPT-3 only (left) and our dual-system model (right). Logically inconsistent lines are written in red text, and are removed from the story-so-far at generation time.

model. We compare with two alternative models (Table 3 in the Appendix): GPT-3 by itself and a dual-system baseline that uses a neural Natural Language Inference (NLI) model as its System 2. The NLI-based dual-system model generates 10 candidates from GPT-3 and selects the candidate with the highest predicted probability of entailment under the NLI model given the context. We use the RoBERTa MNLI model as our off-the-shelf neural NLI model (Liu et al., 2019), which operates as a System 2 that does not use additional problem-specific data or fine-tuning.[2] On 200 held-out tasks, our GPT-3-based "fact extractor" achieves 100% QA accuracy, far exceeding the performance of GPT-3 alone (29.0%) or GPT-3 generation with neural NLI scoring (32.5%; also see Table 3 in the Appendix). These results show that GPT-3 can be made to answer questions successfully when used for parsing with a world model, even when GPT-3 alone does not achieve high QA accuracy.

To test story generation, we use our GPT-3-based System 1 proposal model (few-shot prompted on 10 example stories) to sample a new bAbI story, line-by-line. If a generated utterance is inconsistent with the current state as indicated by the System 2 world model, a new utterance is sampled from System 1 (repeating until a consistent utterance is sampled). Figure 2 shows how the dual-system approach generates stories that mimic the statistical structure of bAbI stories, while remaining logically sound In contrast, GPT-3 alone was not able to maintain logical coherence. In a set of 50 generated stories, all stories required at least one sentence to be resampled to maintain coherence, and over half of the generated sentences (53.1%) were rejected by our System 2 model to maintain logical consistency. These results demonstrate that equipping GPT-3 with a minimal world model produces logically coherent stories that mimic the textural structure of the bAbI domain. In the next section, we apply this approach to mimicking human-generated short stories in natural language.

---

[2]Previous work has used domain-specific entailment/contradiction data to train reranking models (Welleck et al., 2018), however, this requires collecting a dataset of domain-specific entailment and contradiction data.

# 4 Coherent Language Generation - CLUTRR

We apply our dual-system approach to a dataset of natural language using the CLUTRR dataset. CLUTRR contains human-written stories about people and their family relationships (see example in Figure 3). As with bAbI, CLUTRR was originally designed as a Question Answering challenge; instead, we use it to evaluate coherent language generation by querying models to generate complete CLUTRR-style stories or to complete partially-generated stories. Our particular aim is to produce stories with coherent and logically consistent family relationships. As above, our language generation setup consists of pre-trained language models acting as our System 1 proposer, a minimal world model as System 2, and a neural semantic parser (implemented via few-shot GPT-3 prediction) as a bridge between the two systems. We use human judgments to assess whether our neuro-symbolic, dual-system model produces more consistent and coherent stories relative to a baseline.

## 4.1 Model specification

As our System 1 proposal model, we used pre-trained neural models to produce candidate generations one sentence at a time. We experimented with GPT-3 as our System 1 model (which we used above for bAbI), but found generations too unreliable, often outputting the empty string. Instead, we used a BART model (Lewis et al., 2019) that was fine-tuned on the

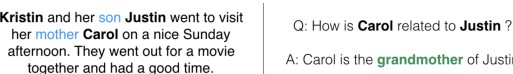

Figure 3: Sample story from the CLUTRR dataset. Each story consists of a sequence of human-generated sentences concerning family relationships. Adapted from Sinha et al. (2019).

CLUTRR training corpus. This model also gives us an opportunity to compare against a best-case neural "single-system" baseline, specifically fine-tuned on story data. To maintain a state of family relations, we use a constraint solver in our "System 2" to encode family relationships (e.g., `child(x, y)`, `spouse(x, z)`) and check that the candidate utterances do not contradict the previous statements (e.g., a person cannot be their own child or married to their sibling). We implemented the world model as a set of logical relations and constraints using the Z3 solver (De Moura & Bjørner, 2008). For instance, we require that the parent of x cannot also be the uncle of x: For all `x, y`, `uncle(x, y)` $\Rightarrow$ `¬child(y, x)`. To check a candidate utterance, we query the solver to determine if the set of constraints is satisfiable or if there is a contradiction. The full set of constraints and other details can be found in the Appendix. We again used GPT-3 as our semantic parser, extracting parses for each candidate utterance via few-shot learning. This parsing approach worked well, even for the natural language in this domain. We observed that parsing with GPT-3 was more successful when the target parse was naturalistic, i.e., *"Bob is Joe's father."* rather than *"father(Bob, Joe)"*. The parsing prompt is reproduced in full in the Appendix.

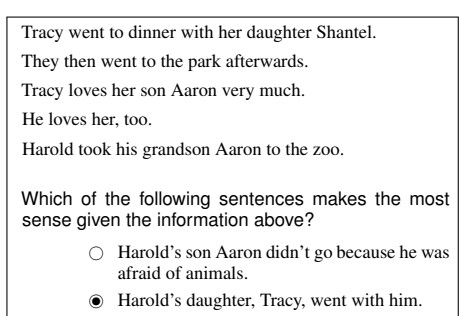

Figure 4: Example trial from CLUTRR human judgement experiment. Participants were instructed to select which of two options makes the most sense given the prompt. One option was generated by the System 1 model only ("single-system"), while the other was generated by the dual-system model.

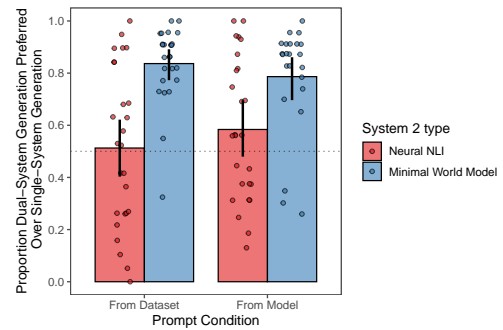

Figure 5: CLUTRR human judgment experiment results. Bars denote proportions of dual-system generations selected as making more sense over single-system generations, in each of four conditions. Error-bars denote bootstrapped 95% confidence intervals of the item means. The points denote means for each individual item in the experiment and are jittered horizontally for clarity.

Table 1: Statistics from CLUTRR story generation. We report the percentage of generations (on both a per-line and per-story basis) for which the System 2 world model did not detect an error. The dual-system model is able to detect many inconsistencies in the neural single-system generations, and most can be corrected by re-sampling new candidates (up to a limit of ten).

| | % w/out error detected (per line) | | % w/out error detected (per story) | |
| | single-system | dual-system | single-system | dual-system |
| | (neural gen. only) | (neural gen.+world model) | (neural gen. only) | (neural gen.+world model) |
| --- | --- | --- | --- | --- |
| prompt from dataset | 82.8 | 97.1 | 60 | 96.1 |
| prompt from model | 71.9 | 96.3 | 36.4 | 93.5 |

## 4.2  Human judgments

We test our dual-system neural generation + world model method in its ability to generate stories that are deemed by naive human participants to be more naturalistic and coherent than those generated from the baseline models. Specifically, we asked participants to select which of two continuations made the most sense to them, where one continuation was generated from the neural model alone (single-system) and the other from a dual-system model (either the world model System 2 or the neural NLI System 2).

**Participants.**  Participants (N = 101) were recruited on the crowd-sourcing platform Prolific and compensated $2 for the task (∼15 minutes, so roughly $8/hour). Participants gave informed consent, and the study was approved by MIT's IRB. 21 participants were excluded for failing an instruction quiz, incorrectly answering more than one of five filler questions, or finishing the task too quickly. The data we collected contains no personally identifiable information or offensive content.

**Procedure.**  Participants began the experiment by reading a set of instructions and answering comprehension questions. On each main trial, participants were shown a prompt consisting of several sentences and were asked to choose which of two possible continuations made the most sense (an example trial is shown in Figure 4). Participants were instructed that if a name appeared multiple times within a trial, then it referred to the same person, whereas if a name appeared across trials, then it was not referring to the same person. For each trial, one continuation option was generated by the neural only single-system baseline, while the other was a dual-system generation. We selected generations from the neural only baseline that were rejected by the System 2 model in order to maximize the differences between the models' generations; thus, human judgments pertain to generations that the models disagreed on. Each participant performed between 20 and 26 trials.

**Materials.**  Participants were randomly assigned to one of four between-participant conditions, which varied according to the kind of prompt and the kind of dual-system model. The prompt was either generated from the model (up to the point of disagreement between System 1 and System 2 models; "Prompts from model" condition) or taken completely from the length 4 CLUTRR systematic generalization test dataset ("Prompts from dataset" condition). To generate prompts for the "from model" condition, we took the first sentence of each story from the CLUTRR test dataset and generated subsequent prompt sentences from the dual-system model; sentences were generated until the two systems disagreed (i.e., System 1 generated a sentence that System 2 rejected), at which point the "rejected sentence" served as the neural only (single-system) baseline generation and the first resampled sentence that System 2 accepted served as the dual-system generation. Prompts were sampled to a maximum length of four sentences. The dual-system model shown to participants used a System 2 based on either our constraint-based "world model" or the neural NLI baseline.

Table 1 catalogs critical statistics from the stimulus generation process. We generated vignettes from the System 1 model and report the percentage of System 1 generations which are deemed correct by the System 2 model.[3] We also report the percentage of generations corrected by the System 2 model (i.e., if System 1 made an error, could System 2 fix it within 10 attempts?). We report these statistics on both a per-story and per-line basis. According to System 2, the System 1 generation model makes a lot of errors (only 36.4% of stories and 71.9% of lines were error-free, in the "from model" condition). In most instances, re-sampling new generations yields stories that, according to

---

[3]For all System 1 generations, we used model temperature of 1.0. For the neural NLI baseline, we used 0.9 probability of contradiction as the cutoff for rejection. Our dual-system model uses a sampling budget of 10 System 1 samples per sentence. Contradictions remaining after 10 samples are considered dual-system errors.

Table 2: Accuracy on gSCAN splits. Models were trained on 5000 examples (only 2.5% of the gSCAN training data). See Appendix Table 4 for additional results.)

| Test split: | single-system[5] | dual-system |
|---|---|---|
| dev | 71.7 | 83.3 |
| random | 57.2 | 74.7 |
| yellow squares | 68.1 | 81.3 |
| red squares | 64.9 | 78.1 |
| novel direction | 0.0 | 0.01 |
| relativity | 41.0 | 53.6 |
| class inference | 68.1 | 76.2 |
| adverb (k=1) | 0.0 | 0.0 |
| adverb to verb | 20.8 | 21.8 |

[3]From Heinze-Deml & Bouchacourt (2020)

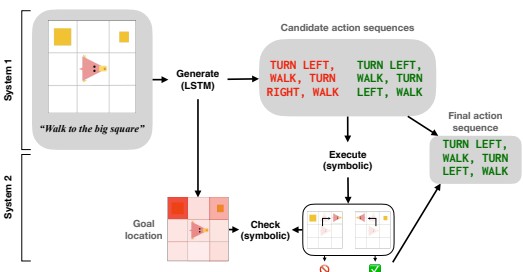

Figure 6: Schematic of our dual-system approach to gSCAN. We train a neural sequence model to predict both a distribution over action sequences, and a distribution over target locations. At test time, we decode candidate action sequences from the model, execute them on the gridworld, and only accept a sequence that brings the agent to the predicted target location (shown in green).

System 2, no longer contain logical errors within a budget of 10 samples (93.5% of stories and 96.3% of lines were error-free, respectively).

**Results.** The human evaluation indicates that System 2 is indeed correcting genuine errors in the stories. As summarized in Figure 5, participants strongly preferred the dual-system neural generation + world model continuations in comparison to the neural only single-system continuations (proportion preferring dual-system = 0.84; bootstrapped 95% confidence interval [0.77, 0.89] and 0.79 [0.77, 0.89] for the "from dataset" and "from model" prompt conditions, respectively). The dual-system approach, however, did not improve generation quality when the System 2 was based on an off-the-shelf neural NLI model (Proportion preferring dual-system = 0.51; [0.40, 0.64] for "from dataset"; 0.58 [0.48, 0.68] for "from model"). Thus, when using a minimal world model, the dual-system approach dramatically improves logical consistency without any need for additional training or fine-tuning. People clearly prefer neuro-symbolic generations from the dual-system model over purely neural generations from a single-system model.[4]

## 5   Grounded Instruction Following

The dual-system approach offers a general-purpose means of improving upon generative, neural sequence models by incorporating logical constraints. To highlight its generality, we examine how the dual-system perspective can be deployed in a very different domain: grounded instruction following. In Heinze-Deml & Bouchacourt (2020), a learned target location predictor was used to increase the accuracy of a neural action sequence generation model. Here, we show how to increase performance further by enforcing consistency between the target location predictor and the action sequence generator in our dual-system framework.

We use the gSCAN benchmark (Ruis et al., 2020), a recently proposed grounded instruction following dataset designed to measure compositional generalization in neural systems. Given an initial gridworld state and an instruction, e.g., "walk to the big square," an agent must predict the sequence of low-level actions which achieve the goal, e.g., "TURN LEFT, WALK, TURN LEFT, WALK" (See Figure 6). The dataset contains several test splits, each testing different aspects of compositional generalization.

Our model builds on Heinze-Deml & Bouchacourt (2020) by using an LSTM to predict the correct action sequence and target location. Given a command $c$ and an initial gridworld state $s$, the neural network defines two distributions: a distribution over action sequences $q_a(a|c, s)$ and a distribution over target grid locations $q_{loc}(l|c, s)$. Heinze-Deml & Bouchacourt (2020) showed that when these distributions share parameters, using location prediction as an auxiliary loss improves the accuracy of the action sequence prediction model. We can further exploit these two models by noticing that when a predicted action sequence is not consistent with a predicted target location, then either the action sequence or the target location must be incorrect. Since the target location is much simpler to predict,

---

[4]We note that the effect size of our approach depends upon the prevalence of disagreements between the single-system and dual-system models. As reported in Table 1, we find that the prevalence of disagreements is at least 14% - 57%, depending on the data regime. Therefore, the model predictions do often differ, resulting in a large overall effect size.

and thus much more likely to be correctly predicted, if a predicted action sequence is not consistent with the predicted target location, then the action sequence is most likely incorrect. Our dual-system framework can use this property to increase action sequence prediction accuracy. Consider the initial state and command in Figure 6. Our model predicts candidate action sequences, and also predicts that the most likely target location is the grid containing the bigger yellow square (highlighted in red). The model then executes the candidate action sequences, and only accepts a sequence which results in the agent standing in the target location.

In the language of our dual-system approach, we treat the distribution over actions $q_a(a|c, s)$ as our **System 1 proposal model**. The distribution over target locations $q_{loc}(l|c, s)$ serves as a **fact extractor** model, which extract a location constraint $l$. As a **minimal world model**, we use a deterministic gridworld execution model $T(a, s_0) \rightarrow s_f$, which takes a state and action and predicts the resulting state. At test time, we first extract the predicted location as $l = \arg\max_{l'} q_{loc}(l'|c)$. We then search through the possible action sequences from $q_a(\cdot|c)$, conditioned on agreement with $l$. In our experiments, we use a sample-based search with a maximum budget of 50 samples. We trained models on random subsets of the gSCAN training set of varying sizes: 5000 datapoints, 8000 datapoints, and 20000 datapoints (2.5%, 4% and 10% of the original training set, respectively).

**Results.** The results show that the System 2 execution model improves performance without the need for any additional training (see Table 2 for results training on 5000 examples). In contrast to the single-system model, the dual-system model allows for sampling many candidate action sequences from the neural network, accepting only consistent sequences. This guess-and-check approach greatly increases the evaluation accuracy, improving upon prior work on gSCAN, particularly in low-data regimes.

# 6  Limitations

In its current form, our approach is most useful in domains where naturalistic, learned generation is necessary and where a small number of mission-critical logical constraints can be explicitly articulated. Our system will be less useful when constraints are more difficult to articulate (e.g., creative domains such as writing poetry) or when there are many constraints, since the minimal world model must be hand-engineered. Enforcing strict constraints may also pose risks: if the constraints are not only logical but cultural, they may be harmful if misapplied. However, these constraints must be articulated explicitly in a symbolic model, and are thus easier to identify and correct.

The current few-shot parsing technique may also suffer from a limited capacity. For more complex domains, the number of examples required to specify the desired parsing behavior may be too large (i.e., they may not fit in the input window) or too complex for a model to perform parsing accurately. While some tasks may not be suitable, the complexity of the world model need not necessarily increase hand-in-hand with the complexity of the application domain. A dual-system model will be most successful when tracking just a few critical variables (e.g., tracking consistency in family relations, as in our experiments, or tracking scheduling constraints when discussing a team plan).

A promising direction for future work is to incorporate learning into the System 2 world model. Currently, the minimal world knowledge that exists in System 2 can be easily modified, but changes must be made by hand. Improvements would come from automatically learning and updating this structured knowledge, possibly by incorporating neuro-symbolic learning techniques (Ellis et al., 2020; Mao et al., 2019), or other neuro-symbolic integration work such as Tsamoura et al. (2021); Michael & Valiant (2008).

Learning could improve our dual-system approach in other ways, e.g., by training a neural module to mimic the actions of a symbolic System 2. The symbolic System 2 judgments could be used as a source of supervision; candidate utterances rejected by the symbolic System 2 model could be used as examples of contradictory sentences, and accepted utterances could be used as examples of non-contradictory statements. This oversight could help train a neural System 2 contradiction-detection model capable of more subtleties than its symbolic counterpart, especially in domains where labeled examples are otherwise unavailable. This approach may also help us understand aspects of human learning, where certain tasks that require slower, logical reasoning can be habitualized over time and tackled by faster, more intuitive reasoning.

Recent work (Li et al., 2021) has shown that large pre-trained neural models learn to approximately represent certain types of structured semantic information. However, it is not yet clear how representational fidelity translates to logical coherence during generative tasks. Our current approach allows us to explicitly fix logical errors in generation, which may ultimately be caused by representational errors. Understanding how we might leverage our approach to improve the representation of structured knowledge within neural models is a promising direction for future work, which could lead to increased generation consistency and coherence.

## 7  Conclusion

Inspired by dual process theories from cognitive science, we combine the respective strengths of neural and symbolic approaches to build more robust models that can more effectively incorporate domain knowledge. For language generation, we showed that equipping neural generation with a minimal symbolic world model increased language coherence and consistency. For grounded instruction following, we showed that requiring test-time consistency between predicted action sequences and goal locations led to improved performance, especially in low-data regimes. Our neuro-symbolic approach can readily be applied to other domains and types of prior knowledge, as a lightweight way of improving the coherence and consistency of powerful neural sequence models.

This paper just scratches the surface of how structured knowledge can make neural systems more robust; we hope to inspire further work into neuro-symbolic systems which possess the robustness and commonsense necessary for human-level intelligence.

### Acknowledgments

We thank Laura Ruis, Jacob Andreas, Yewen (Evan) Pu, Joe O'Connor and Guy Davidson for helpful comments on an earlier version of this manuscript. MN is supported by a NSF Graduate Research Fellowship.

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
