# A    Additional CRT experiments

Figure 8 shows additional cognitive reflection test (CRT) experiments using GPT-3. For each question type, the original is displayed along with a table containing the responses for several variants. Prompt text is in black, and response text is in color (green for correct, red for incorrect). Since it is unknown whether the CRT tasks are within the GPT-3 training set, we test several conceptually similar variants. All experiments were performed with the "davinci" model using temperature 0 and the newline token set as the stop sequence. For each question, GPT-3 often (but not always) makes the same mistakes as humans, even when the numbers are modified from the original CRT task.

# B    Experimental Details

For all experiments using GPT-3, we used the largest available "davinci" model. All other models were implemented in PyTorch. All testing and training was performed on one Nvidia GTX 1080 Ti GPU. Language generation experiments generally took less than 4 hours to run, and gSCAN experiments took less than 48 hours.

## B.1    bAbI

Table 3: Question answering results on bAbI.

| Model | Acc. |
| --- | --- |
| GPT-3 generation only | 29.0% |
| GPT-3 generation, neural NLI scoring | 32.5% |
| parsing (via GPT-3) + Sys 2 world model | 100% |

The full text of the parsing prompts for the bAbI domain can be found in Figure 9. QA results are shown in Table 3.

**Minimal world model.**    The minimal world model for bAbI is implemented via simple Python code and performs three functions:

1. Tracks the people, objects and locations which have been mentioned so far.
2. Modifies the world state changes as a result of parsed actions.
3. Checks if the candidate action violates the current world state, as defined by (1) and (2).

Tracking the people, objects and locations is performed by maintaining a lookup table which maps the string representing a name (e.g., *'football'*) to the Python object that represents the corresponding person, object or location. The Python objects inherit from one of three small classes, `Person`, `Location`, or `Obj`. When a new person, object or location is referenced, a new corresponding Python object is initialized and added to the lookup table. The logic for possible actions (in our experiments, `pickup`, `go`, and `drop`) are implemented to carry out the intended action. For instance, `pickup(person, obj)` adds `obj` to the inventory of `person`. Likewise, `go(person, location)` changes the location of `person` and `person`'s inventory to `location`. Each action checks whether the current world state satisfies necessary preconditions for the action. If the current world state violates the action preconditions, an error is thrown, and the candidate action is rejected. For example, if the location of `obj` (`obj.location`) has been specified and is not the same as the location of `person`, then `pickup(person, obj)` will fail.

The full world model used for bAbI (including code for interpreting the output of the GPT-3 fact extractor) consists of fewer than 200 lines of code, and can be found in `worldModel.py`.

## B.2    CLUTRR

For our CLUTRR experiments, we used a BART-base model (Lewis et al., 2019), which was fine-tuned on the "systematic generalization" training corpus with story lengths of 2-4 sentences from the CLUTRR dataset (folder `data_db9b8f04` from the dataset found at

Table 4: Results for each gSCAN split, showing exact match accuracy. The dual-system model outperforms baselines in nearly all test splits, and is especially beneficial in the low-data regime.

| test split: | 5000 examples | | 8000 examples | | 20000 examples | |
|---|---|---|---|---|---|---|
| | single-system[6] | dual-system | single-system | dual-system | single-system | dual-system |
| dev | 71.7 | 83.3 | 81.0 | 90.7 | 92.6 | 96.9 |
| random | 57.2 | 74.7 | 69.1 | 84.6 | 88.6 | 95.2 |
| yellow squares | 68.1 | 81.3 | 77.2 | 89.3 | 89.1 | 94.2 |
| red squares | 64.9 | 78.1 | 76.3 | 88.1 | 87.8 | 93.2 |
| novel direction | 0.0 | 0.01 | 0.0 | 0.1 | 0.0 | 0.02 |
| relativity | 41.0 | 53.6 | 40.0 | 50.1 | 62.5 | 68.9 |
| class inference | 68.1 | 76.2 | 78.1 | 87.6 | 89.4 | 96.6 |
| adverb (k=1) | 0.0 | 0.0 | 0.0 | 0.0 | 0.0 | 0.0 |
| adverb to verb | 20.8 | 21.8 | 20.2 | 20.6 | 19.9 | 21.4 |

[4]From Heinze-Deml & Bouchacourt (2020)

github.com/facebookresearch/clutrr). BART models were retrieved from Hugging Face and fine-tuned with the Hugging Face default AdamW (Loshchilov & Hutter, 2017) optimizer with a learning rate of 1e-5 and dropout of 0.1. The full text of the parsing prompt for the CLUTRR domain is shown in Figure 10. Statistics for the CLUTRR story generation for both the symbolic world model and neural NLI System 2 models are shown in Table 5. Additional trials from the human judgement experiment are shown in Figure 11.

We also explored how the sampling budget affects story generation statistics for the symbolic world model. We found that using a budget of 5 samples instead of 10 has a relatively small effect on overall results: in the "Prompts from dataset" condition, using a sampling budget of 5 reduced the percentage of error-free lines generated by the dual-system model to 96.% (down from 97.1%), and the percentage of error-free stories to 93.5% (down from 96.1%). In the "Prompts from model" condition, the percentage of error-free lines is 94.6% (down from 96.3%), and the percentage of error-free stories is 88.3% (down from 93.5%).

The family relation constraints used for the world model can be found in Figure 7. The code implemented these constraints using the Z3 solver can be found in clutrrZ3.py. All constraints are gender neutral, and read left to right, e.g., child$(x, y) \rightarrow$ *the child of x is y*. We follow the notation used in Sinha et al. (2019), and use the term grand for *grandchild*, un for *aunt/uncle*, etc., and use the prefix inv_ for inverses of already-defined terms.

We accept the following terms as outputs of the GPT-3-based parsing system, mapping them to the correct constraint: *'spouse', 'husband', 'wife', 'parent', 'grandchild', 'granddaughter', 'grandson', 'grandparent', 'grandmother', 'grandfather', 'father', 'mother', 'uncle', 'aunt', 'nephew', 'niece', 'sister', 'brother', 'daughter', 'son', 'daughter in law', 'son in law', 'mother in law', 'father in law', 'mother-in-law', 'father-in-law', 'daughter-in-law', 'son-in-law.'* All other terms output from the parsing system (e.g., 'Mark is Mary's *friend*') are ignored and do not lead to the addition of new constraints to the current world state.

## B.3 gSCAN

Table 4 shows the accuracy results for gSCAN on training sets of size 5000, 8000, and 20000.

**Architecture** Our model architecture is identical to the "Both" attentional variant from Heinze-Deml & Bouchacourt (2020). In this model, a BiLSTM encoder encodes the instruction sequence and a CNN encoder encodes the initial gridworld state. To predict a target location, attention-weighted gridworld state encodings (which also attend over the instruction sequence encoding) are passed to a linear classification layer, which predicts softmax scores for each gridworld position. To predict an action sequence, an LSTM decodes the action sequence based on attention weighted instruction encodings and attention weighted gridworld encodings. The gridworld encoding vectors are additionally weighted by the softmax scores from the target location prediction layer before being fed to the LSTM decoder attention mechanism. We use the same hyperparameters as Heinze-Deml &

Relations:
$\forall x, y, z.\texttt{child}(x, z) \wedge \texttt{child}(z, y) \implies \texttt{grand}(x, y)$
$\forall x, y, z.\texttt{grand}(x, z) \wedge \texttt{sibling}(z, y) \implies \texttt{grand}(x, y)$
$\forall x, y, z.\texttt{inv\_child}(x, z) \wedge \texttt{inv\_child}(z, y) \implies \texttt{inv\_grand}(x, y)$
$\forall x, y, z.\texttt{sibling}(x, z) \wedge \texttt{inv\_grand}(z, y) \implies \texttt{inv\_grand}(x, y)$
$\forall x, y, z.\texttt{child}(x, z) \wedge \texttt{sibling}(z, y) \implies \texttt{child}(x, y)$
$\forall x, y, z.\texttt{SO}(x, z) \wedge \texttt{child}(z, y) \implies \texttt{child}(x, y)$
$\forall x, y, z.\texttt{sibling}(x, z) \wedge \texttt{inv\_child}(z, y) \implies \texttt{inv\_child}(x, y)$
$\forall x, y, z.\texttt{child}(x, z) \wedge \texttt{inv\_grand}(z, y) \implies \texttt{inv\_child}(x, y)$
$\forall x, y, z.x \neq y \wedge \texttt{inv\_child}(x, z) \wedge \texttt{child}(z, y) \implies \texttt{sibling}(x, y)$
$\forall x, y, z.\texttt{child}(x, z) \wedge \texttt{SO}(z, y) \implies \texttt{in\_law}(x, y)$
$\forall x, y, z.\texttt{SO}(x, z) \wedge \texttt{inv\_child}(z, y) \implies \texttt{inv\_in\_law}(x, y)$

$\forall x, y, z.\texttt{sibling}(x, z) \wedge \texttt{child}(z, y) \implies \texttt{inv\_un}(x, y)$
$\forall x, y, z.\texttt{inv\_child}(x, z) \wedge \texttt{sibling}(z, y) \implies \texttt{un}(x, y)$

Inverses:
$\forall x, y.\texttt{child}(x, y) \iff \texttt{inv\_child}(y, x)$
$\forall x, y.\texttt{inv\_in\_law}(x, y) \iff \texttt{in\_law}(y, x)$
$\forall x, y.\texttt{inv\_grand}(x, y) \iff \texttt{grand}(y, x)$
$\forall x, y.\texttt{inv\_un}(x, y) \iff \texttt{un}(y, x)$

Symmetric rules:
$\forall x, y.\texttt{sibling}(x, y) \iff \texttt{sibling}(y, x)$
$\forall x, y.\texttt{SO}(x, y) \iff \texttt{SO}(y, x)$

Definition of an ancestor:
$\forall x, y.\texttt{child}(x, y) \implies \texttt{ancestor}(y, x)$
$\forall x, y.\texttt{grand}(x, y) \implies \texttt{ancestor}(y, x)$

Sibling is transitive:
$\forall x, y, z.x \neq z \wedge \texttt{sibling}(x, y) \wedge \texttt{sibling}(y, z) \implies \texttt{sibling}(x, z)$

You can't be your own ancestor:
$\forall x, y.\texttt{ancestor}(x, y) \implies \neg\texttt{ancestor}(y, x)$ Ancestor transitivity:
$\forall x, y, z.\texttt{ancestor}(x, y) \wedge \texttt{ancestor}(y, z) \implies \texttt{ancestor}(x, z)$
You can't be your own sibling:
$\forall x.\neg\texttt{sibling}(x, x)$
Inverse of aunt/uncle:
$\forall x, y.\texttt{inv\_un}(x, y) \implies (\exists z.\texttt{sibling}(x, z) \wedge \texttt{child}(z, y))$
Inverse of sibling:
$\forall x, y.\texttt{sibling}(x, y) \implies (\exists z.\texttt{child}(z, x) \wedge \texttt{child}(z, y))$
Parents can't be aunts/uncles:
$\forall x, y.\texttt{un}(x, y) \implies \neg\texttt{inv\_child}(x, y)$ $\forall x, y.\texttt{inv\_child}(x, y) \implies \neg\texttt{un}(x, y)$

Figure 7: Family constraints for CLUTRR world model. Based on constraints used in Sinha et al. (2019). All constraints are gender neutral, and read as left to right, e.g., $\texttt{child}(x, y) \rightarrow$ *the child of x is y*.

Table 5: Statistics from CLUTRR story generation.

| | neural NLI System 2 | | | | minimal world model System 2 | | | |
| | % w/out error detected (per line) | | per story | | per line | | per story | |
| | single-system | dual-system | single-system | dual-system | single-system | dual-system | single-system | dual-system |
| --- | --- | --- | --- | --- | --- | --- | --- | --- |
| prompt from model | 73.3 | 98.4 | 40.2 | 94.8 | 71.9 | 96.3 | 36.4 | 93.5 |
| prompt from dataset | 82.9 | 100 | 57.1 | 100 | 82.8 | 97.1 | 60 | 96.1 |

Bouchacourt (2020), including a CNN dropout rate of 0.1, a dropout rate of 0.3, an auxiliary task weight of 0.3, and LSTM sizes of 100. See Heinze-Deml & Bouchacourt (2020) for more details.

**Test-time search.** At test time, the neural only single-system baseline from Heinze-Deml & Bouchacourt (2020) performs greedy decoding. To directly compare to the single-system model, the dual-system model performed greedy decoding to produce the first candidate action sequence. If this candidate action sequence failed the consistency check, the model proceeded with a sample-based search, as described above, with a sampling budget of 50 samples.

**Consistency check.** The target location prediction in Heinze-Deml & Bouchacourt (2020) predicts the initial location of the target object. However, to successfully complete some of the actions in the gSCAN domain, such as "push" and "pull", the agent must move the target object to a different grid. Therefore, the final grid of the agent is not always the same as the target location. To account for this, the consistency check required only that the agent *passes through* the target location and does not move outside the bounds of the gridworld. This is a strictly less stringent constraint than requiring that the agent's final location matches the target location; nevertheless, we see that this constraint is sufficient to achieve significant accuracy gains.

A ball and a bat cost **$1.10.**
The bat costs one dollar more than the ball.
How much does the ball cost?
Answer: 10 cents

| Total cost | Response |
|---|---|
| $1.10 | 10 cents |
| $1.20 | 20 cents |
| $1.30 | $0.30 |
| $1.70 | $0.70 |

If it takes **5** machines **5** minutes to make **5** widgets, how long would it take 100 machines to make 100 widgets?
Answer: 5 minutes.

| Value | Response |
|---|---|
| 5 | 5 minutes |
| 6 | 6 minutes |
| 7 | 7 minutes |
| 8 | 8 minutes |
| 9 | 9 minutes |
| 10 | 10 minutes |
| 11 | 100 minutes |
| 12 | 100 minutes |
| 13 | 100 minutes |

In a lake, there is a patch of lily pads.
Every day, the patch doubles in size.
If it takes **48** days for the patch to cover the entire lake, how long would it take for the patch to cover half of the lake?
Answer: 24 days

| # of days | Response |
|---|---|
| 8 | (no output) |
| 20 | 10 days |
| 24 | 12 days |
| 32 | (no output) |
| 25 | (no output) |
| 36 | (no output) |
| 48 | 24 days |
| 100 | 50 days |

Figure 8: GPT-3 responses to CRT problems and variants. Correct answers shown in green, and incorrect answers shown in red. "(no output)" indicates that the model produced the newline token (set as the stop sequence) and did not produce an output.

```
Please parse the following statements into commands. The available commands are pickup, drop, and go.
Sentence: Max journeyed to the bathroom.
Semantic parse: go(Max, bathroom)

Sentence: Mary grabbed the football there.
Semantic parse: pickup(Mary, football)

Sentence: Bob picked up the apple.
Semantic parse: pickup(Bob, apple)

Sentence: Susan dropped the milk.
Semantic parse: drop(Susan, milk)

Sentence: Bob got the football there.
Semantic parse: pickup(Bob, football)

Sentence: Max left the cup.
Semantic parse: drop(Max, cup)

Sentence: Kevin put down the pie there.
Semantic parse: drop(Kevin, pie)

Sentence: John took the football there.
Semantic parse: pickup(John, football)

Sentence:
```

---

```
Please parse the following questions into queries using queryObjLoc:
Question: Where is the toothbrush?
Semantic parse: queryObjLoc(toothbrush)

Question: Where is the milk?
Semantic parse: queryObjLoc(milk)

Question: Where is the apple?
Semantic parse: queryObjLoc(apple)

Question: Where is the football?
Semantic parse: queryObjLoc(football)

Question:
```

Figure 9: Semantic parsing prompts for bAbI domain. Top: Statement semantic parsing prompt. Bottom: Question semantic parsing prompt.

The following sentences contain people and their family relationships. Please parse each sentence into family relationships.
The available relationships are sibling, parent, child, grandchild, uncle, spouse.
If a sentence has no relationship, say "None".

Sentence: Michael's sister, Mary, was crying, so he told her a joke.
Semantic parse: Mary is Michael's sister.

Sentence: Joshua's son, Clarence, loves trains.
Semantic parse: Clarence is Joshua's child.

Sentence: David is very lucky to have a husband who adores her and treats her like a queen.
Semantic parse: None

Sentence: Lillian is married to Thomas and when she was 24, the couple welcomed April into the world.
Semantic parse: Thomas is Lillian's spouse. April is Lillian's child.

Sentence: Bobby does n't like his grandfather James.
Semantic parse: Bobby is James's grandchild.

Sentence: He loved spending time with her, and she loved it too.
Semantic Parse: None

Sentence: Jerry asked his father George if he could borrow some money.
Semantic parse: Jerry is George's child

Sentence: Robert and his brother Louis watch Robert's daughter Michelle in her school play.
Semantic parse: Louis is Robert's sibling. Michelle is Robert's child.

Sentence: Bernardo got a cone and Antonio got a sundae.
Semantic parse: None

Sentence: They had a wonderful time.
Semantic parse: None

Sentence: Mary was playing in the sandbox with her brother Dennis.
Semantic parse: Mary is Dennis's sibling.

Sentence:
Semantic parse: None

Sentence: David is very lucky to have a husband who adores her and treats her like a queen.
Semantic parse: None

Sentence: Dennis id Amy's only child.
Semantic parse: Dennis is Amy's child.

Sentence: Michael laughed, and felt better.
Semantic parse: None

Sentence: Angelica ca n't wait to see her favorite aunt Tracy.
Semantic parse: Tracy is Angelica's aunt.

Sentence: Marie does n't like having to babysit her younger brother, Pedro.
Semantic parse: Pedro is Marie's sibling.

Sentence:

Figure 10: Semantic parsing prompt for CLUTRR domain.

Marie and her husband Robert like to go fishing on the weekends.

Marie's son Allan doesn't go because he hates fishing.

Which of the following sentences makes the most sense given the information above?

- ○ Allan and his uncle Robert went to the park.
- ◉ Allan and his aunt, Beverly, went to Disney World.

---

Robert and his brother Antonio played harmonicas together.

Robert's daughter, Elsie, asked him to play with her.

Which of the following sentences makes the most sense given the information above?

- ○ Elsie doesn't like having to babysit her younger brother, Antonio.
- ◉ Elsie was playing hide-and-seek with her sister Tracy.

---

Tracy went to dinner with her daughter Shantel.

Shantel's daughter, Lizzie, asked her mom to read her a story.

Which of the following sentences makes the most sense given the information above?

- ○ Lizzie was playing hide-and-seek with her sister Tracy.
- ◉ Lizzie loves hanging out with her uncle Antonio.

Figure 11: Additional trials from CLUTRR human judgement experiment ("prompt from model" condition). Selected option was generated by the dual-system model, and the other option was generated by the neural only single-system model.