# OpenReview forum: "Improving Coherence and Consistency in Neural Sequence Models with Dual-System, Neuro-Symbolic Reasoning"
_NeurIPS.cc/2021/Conference — NeurIPS 2021 Poster_

### Official Review · Reviewer_2tE8 · 2021-06-27

**Rating:** 5
**Confidence:** 4

**Summary:**

The paper proposes a neuro-symbolic model to improve coherence and consistency in neural language generation models. In particular, the authors use a neural language model to generate samples, and reject invalid ones using a world model built on context sentences and the parsed results on the generated sentences, and resample if necessary. In a set of experiments (bAbI, CLUTRR, and gSCAN), the authors show that the proposed system could improve coherence and consistency in the generated sequences.

**Limitations And Societal Impact:**

Yes.

**Main Review:**

Pros:
* I actually quite like the idea of combining neural models and logic systems to improve performance, especially in achieving greater human-likeness. The proposed idea, *i.e.*, using a symbolic workflow to reject and resample, is quite intuitive, and the reported results are encouraging.
* I also like how the authors present the experiments, detailing the experimental protocol, the human study procedure, and a lot of other small design choices.

Cons:
* The way to combine the two systems is too simple: without further integration, success of the separate processes is completely dependent on luck. You could neither revise / correct the errors, nor bound the number of trials. If you are very unlucky, you might get trapped in a region with no valid generations, or you need a very large number of resamples to get a valid one.
* The world model requires ad-hoc design. This could be a potential issue in scalability. When a problem domain becomes relatively large, it would be really hard to track everything.
* The parsing process (and potentially the world model building process) involves additional neural processing. How to ensure coherence and consistency in these parts? Will it be a chicken-and-egg problem? What if the parsing / world-model-building process gives you wrong answers?

In summary, I like the idea and the general methodology presented in this work. However, it also bears some limitations from Good-Old-Fashion-AI, most of them discussed in the Limitation section. Despite of that, I think the general methodology should be encouraged but I would also love to sync with other reviewers just in case I miss some points / issues.

Post rebuttal: I have read the authors' rebuttal and other reviews. The authors basically agree that the limitations exist and they don't have a very nice way to solve them for now. Therefore, I keep my initial rating.

**Time Spent Reviewing:**

1.5

---

> ### Author Response · Authors · 2021-08-10
> **Response to Reviewer 2tE8**
>
> Thank you very much for your review.
> - **Simple sampling-based inference:** Although the sampling-based method by which we combine our two models is simple, we have shown empirically that it leads to gains in consistency. We also imagine that more sophisticated ways to combine the systems could be an area of future work (see the Limitations section).
> - **World model:** Our approach does require designing the world model for each domain. However, we do note that, for our approach to succeed, it does not necessarily require tracking all of the important state information; in the gSCAN domain, our dual-system model increases performance over the neural baseline, even though the System 2 model only encodes the goal location, which is just one aspect of the information needed to solve the problem (see the explanation in response to Reviewer Ms3t). Again, we view this as a strength of our approach; it can be applicable in domains where a full symbolic treatment is not feasible.
> - As discussed in the limitations, we believe that integrating this work with approaches for learning symbolic models could be a promising area for future research (Ellis et al. 2021, Michael and Valiant, 2008, and others). This may help address the scalability issues you mention, as well as address issues concerning consistency in the parsing process by, for example, learning to use effective parsing prompts.
>
> **References:**
> Ellis, K., Wong, C., Nye, M., Sablé-Meyer, M., Morales, L., Hewitt, L., ... & Tenenbaum, J. B. (2021). DreamCoder: bootstrapping inductive program synthesis with wake-sleep library learning. In Proceedings of the 42nd ACM SIGPLAN International Conference on Programming Language Design and Implementation (pp. 835-850).
>
> Michael, L., & Valiant, L. G. (2008). A First Experimental Demonstration of Massive Knowledge Infusion. In KR (pp. 378-389).

---

### Official Review · Reviewer_CnYk · 2021-07-09

**Rating:** 4
**Confidence:** 4

**Summary:**

This paper tries to improve neural network reasoning by combing the deep-learning-based System 1 and logical-reasoning-based System 2. The proposed approach uses an attached logical reasoning module to check the consistency of the predicted text by GPT-3. In order to connect neural NLP with logical reasoning, it pre-trains a GPT-3/BART model to extract ground atoms from both the input and predicted natural language sentences, and then a domain-specific language (DSL) based reasoning module searches in the pre-defined knowledge base to check whether the extracted groundings are logically consistent. The top-ranked consistent result is returned as the final output.

**Limitations And Societal Impact:**

1. The use of GPT-3 limits the applicability of the presented approach.
2. The proposed system can only make inferences and does not learn.
3. The paper is missing some very related works.

**Main Review:**

Separating end-to-end neural reasoning into multiple-stage processing that involves both neural computing and logical calculi seems to be a promising direction for AI. This work presents a straightforward and effective way to address natural language reasoning tasks. The main idea in this paper is to extract logical atoms from text and then uses a pre-defined knowledge base to constrain the predicted result. The advantage of this work is that the pure deep learning-based prediction can now be filtered by the symbolic constraints. The symbols extracted by the pre-trained GPT-3 model has shown their effectiveness empirically in the presented experiments.

There are few points that are not clear to me. The symbol extraction requires both supervised pre-training and a well-defined DSL knowledge base. The paper, at least in its main text, does not include any information about how much-supervised data is required and how to design such a  DSL knowledge base in general, e.g., for tasks other than the ones in this paper. Thus, I cannot tell whether the proposed approach is scalable and general enough in practice. Another problem is that this work heavily depends on GPT-3, which is not an open AI model. Even for GPT-3, its performance on the CLUTRR dataset is still questionable, so I wonder how applicable is the proposed approach. The logical reasoning module is only used for consistency checking, while the System 2 for "logical reasoning" is actually done by GPT-3, which is not "logical" at all. Furthermore, the "reasoning" cannot be improved as the System 2 GPT-3 is not updated anymore.

This paper has clearly missed many references, especially those neuro-symbolic systems rooted in the classic symbolic AI. Many works in this area have proposed to improve deep learning via symbolic consistency checked by a systematic logical inference engine. For example:
- R. Manhaeve, et al., Deepproblog: Neural probabilistic logic programming, NeurIPS. 2018.
- W. Dai, et al., Bridging machine learning and logical reasoning by abductive learning, NeurIPS. 2019.
- R. Evans, et al., Making sense of raw input. Artificial Intelligence, 299:103521, 2021.

The above related works aim at training neural networks/symbolic models by including a symbolic System 2 module without (strong & data hungry) pre-training, which I personally think are stronger than this work. Meanwhile, similar ideas (i.e., using symbolic rules to constrain the machine learning models' outputs to get reasonable predictions) have been explored even earlier. For example:
- I. Tsochantaridis, et al.,. Support Vector Learning for Interdependent and Structured Output Spaces, ICML, 2004.
- S. Mei, et al., Robust RegBayes: Selectively Incorporating First-Order Logic Domain Knowledge into Bayesian Models, ICML, 2014.

Detailed questions:
- Line 77, "few-shot learning approach" and "requires no additional training", which one is correct? From what I can tell, the paper does require some supervision for pre-training, which is still "training".
- Does the GPT-3 training data contain bAbI text?
- How to ensure GPT-3 outputs the predicate logic format defined by the DSL in this paper?
- How many pre-train data for semantic parsing?
- Line 166: "could easily be applied to other tasks". How easy is it? does it need pre-training or the model learned here can be directly adapted to other domains?
- The compared NLI does not uses external logic constraints and pre-training.
- Why not fine-tune GPT-3 on CLUTRR directly instead of using another BART model?


**Time Spent Reviewing:**

2

---

> ### Author Response · Authors · 2021-08-10
> **Response to Reviewer CnYk**
>
> Thank you for your detailed review.
> - **GPT-3 few-shot parsing:** We would like to highlight a potential misunderstanding: In the bAbI and CLUTRR text generation domains, the parsing is performed by GPT-3. GPT-3 is a 170+ billion parameter neural language model, trained and hosted by OpenAI, for which we do not have access to the weights or training data. We cannot fine-tune GPT-3, and can only run sampling-based inference queries. Therefore, the GPT-3 parser is not fine-tuned on any additional data. As described in lines 158-169, our fact extraction procedure involves “few-shot prompting” (Brown et al, 2020), where a small number of examples of the desired input behavior are given in the prompt, along with the utterance in question. The subsequent GPT-3 generation is interpreted as the target parse.  Specifically, we ran inference using the generation prompts shown in Appendix Figs 9 and 10.
> - **Related work:** Thank you for the additional references to related work. Our system differs from work such DeepProbLog in that we use neural language models trained on either self-supervised or supervised data to produce rich generation candidates, and a lightweight symbolic module to reject candidates, without additional neural model training. We will revise the draft to make this more clear, and we will add references and discussion of the papers mentioned.
>
>
> **Responses to detailed questions:**
> - **“Line 77, "few-shot learning approach" and "requires no additional training", which one is correct? From what I can tell, the paper does require some supervision for pre-training, which is still "training".”** In this sentence, we were referring specifically to the GPT-3 based parser used in the text generation CLUTRR and bAbI domains, which, as discussed above, we do not fine-tune. We can clarify this in our revision.
> - **“Does the GPT-3 training data contain bAbI text?”** That’s a great question. Unfortunately we do not have access to the official dataset used to train GPT-3. It’s also worth pointing out that even if GPT-3 was trained on the bAbI data, our results show that its bAbI generation performance is far from perfect.
> - **“How to ensure GPT-3 outputs the predicate logic format defined by the DSL in this paper?”** We did not take any steps to ensure that the output generated by GPT-3 conformed to our target format besides using the prompts shown in Appendix Figs 9 and 10 for bAbI and CLUTRR experiments, respectively. GPT-3 was able to output well-formed responses. As discussed on line 260, we did find in the CLUTRR domain that parsing was more successful when the target parse was naturalistic, i.e., “Bob is Joe’s father.” rather than “father(Bob, Joe)”.
> - **“How many pre-train data for semantic parsing?”** As discussed above, the GPT-3 based semantic parser did not use any additional fine-tuning for semantic parsing. Figure 10 in the appendix shows the entire inference prompt used for semantic parsing in the CLUTRR domain, which contains 17 examples of semantic parsing.
> - **“Line 166: "could easily be applied to other tasks". How easy is it? does it need pre-training or the model learned here can be directly adapted to other domains?”** Since we are using the GPT-3 model without any fine-tuning, we imagine it can also be used without fine-tuning for other domains. As mentioned on line 167, this is also shown to work well in Shin et al. (2021).
> - **“The compared NLI does not uses external logic constraints and pre-training.”** Are you able to clarify this question? We don’t understand what it means. Thank you!
> - **“Why not fine-tune GPT-3 on CLUTRR directly instead of using another BART model?”** As discussed above, we do not have access to the GPT-3 weights and are not able to fine-tune it. We highlight this as a motivation for our approach: we have shown that we can achieve increased consistency using neural models without being able to additionally fine-tune them, which we suspect will be an increasingly common practical limitation.
>
> **References:**
> Brown, T. B., Mann, B., Ryder, N., Subbiah, M., Kaplan, J., Dhariwal, P., Neelakantan, A., Shyam, P., Sastry, G., Askell, A. & others (2020). Language models are few-shot learners. NeurIPS.
>
> Shin, R., Lin, C.H., Thomson, S., Chen, C., Roy, S., Platanios, E.A., Pauls, A., Klein, D., Eisner, J., & Durme, B.V. (2021). Constrained Language Models Yield Few-Shot Semantic Parsers. ArXiv, abs/2104.08768.

---

> > ### Comment · Reviewer_CnYk · 2021-08-26
> > **Thanks for the feedback**
> >
> > I appreciate the authors' feedbacks, which has answered my major concern about using the pre-trained GPT-3. I agree that using those foundation models that pre-trained from a large number of collected training data can make System 2 reasoning tasks easier. However, a fixed pre-train model also limits the model's generalization potential, which harms a crucial benefit of System 2 reasoning, out-of-distribution generalization, or extrapolation.
> >
> > This also brings another concern to me. The authors have complained a lot in their feedback about they have no access to GPT-3. Hence, these large pre-trained models like GPT-3 are controlled by big companies, making our model's performance depend on them would be unwise because this harms the openness of scientific research since we cannot do anything about it.
> >
> > Overall, technically I think borrowing the power of GPT-3 is very useful, but I agree with the other reviewers that it also loses novelty. Lacking the ability to fine-tune the System 1 model also makes the contribution weaker comparing to those referred papers in my review. So I would like to keep my rating.

---

### Official Review · Reviewer_Ms3t · 2021-07-15

**Rating:** 3
**Confidence:** 4

**Summary:**

The authors take some natural language text `X`, and use GPT-3 to output a machine-readable piece of text representing the world state `S`. Next, they get a model to output a set of candidate next sentences `Yᵢ` for `X`. Another GPT-3 model is used to output a machine-readable state-change `Δᵢ`. Symbolic methods are used (e.g., Z3 with hard-coded rules) to determine whether `S` and `Δᵢ` are consistent, and finally the `Yᵢ` that are consistent are put forward as states. The claim is that this will produce better, logically coherent text.

**Ethical Concerns:**

The only place where an ethical issue may arise is in the use of humans to assess the output. But the authors state in the paper that the participants were renumerated and gave informed consent so there are no ethical issues.

**Limitations And Societal Impact:**

Limitations are addressed in the paper:

- the constraints must be constructed manually (can’t be learned) and so you can’t scale or change domains easily;

- because the formal translation is done with a language model, there is limited capacity. So it might only work with a few variables.

Societal impact is addressed: enforcing the constraints on ’System 2’ might include cultural constraints rather than logical constraints (e.g., forbids poetic language).
The authors also claim that having an explicit set of rules in ’system 2’ makes it easier to find these harmful assumptions.

**Main Review:**

The paper is appropriate for NeurIPS. The paper is organised and presented well.

# Validity:

Overall, there are not that many technical parts of the paper in which I could check for validity. On a less formal level, I don't quite buy that 'using Z3' is the same as what Kahneman means by 'System 2’.
In the human-judgment experiment, I take issue with the 'single-system' sample being a sample which was _rejected_ by 'System 2'. This is not comparing 'System 1 + 2' with 'System 1'; it is comparing 'System 1 + 2' with 'System 1 - 2'.

I also do not think the experiment in section 5 is valid:  why does the LSTM need to produce the sequence of command-words if it is producing a heat map on where the goal is and a symbolic check that the sequence of words finishes on the goal? Once you are doing the symbolic check and know the goal position you can easily generate the command-word list symbolically. This is not a valid experiment for testing the claim that a system 2 helps with the prediction, if system 2 doesn't need system 1.

# References:

I think that including all the cognitive science is a distraction from the core of the paper.
The results of using a 2-system approach should stand on their own.
It also reminds me of the general idea of 'autoformalization' as described, e.g., in Szegedy at CICM 2020.

# Contribution:

The contribution is to use generate-and-rerank, but with the reranking performed by composing hybrid language model + symbolic checker. This is not a significant enough contribution and the problems tackled are easy. The symbolic checker's rules had to be manually coded and so I am not convinced that this approach is going to scale.

The notion of having a two-system-style model is also very common.

There has also already been a lot of work on using GPT to generate machine-readable text.

There has also already been work on using language-model-generated, machine-readable text to guide a search, for example, for lemma selection in automated theorem proving. So to me it is not a great leap to join these pieces together. The challenge in this space is in how to remove the human dictating the rules of system 2.

The experiments do not compare the performance of their system to any other techniques, e.g., Welleck 2018. It's only compared to a baseline (which I argue is unfair for both experiments).

# Details

- This sounds exactly like generate-and-rerank, so how is yours different to the related work?

- Related work should include a few sentences on using language models for logical reasoning; e.g., original GPT3 paper, Lample ICLR 2020.

- _Thinking Fast and Slow_ is not an appropriate citation. Surely Kahneman has a specific cogsci paper or citation on the same specific point you are trying to make? For example, the ball+bat study is an actual study that can be cited.

- Lots of these citations are arxiv preprints. I know that the field is moving so fast that you need to cite preprints. But in some cases there is a published version of the work, e.g., the preprint (Lake, Murphy 2020) says "In press at Psychological Review". Maybe at the time of writing, this hadn't been published yet, but really you should be citing published, peer-reviewed papers if you are using the citation to support an argument.


**Time Spent Reviewing:**

5

---

> ### Author Response · Authors · 2021-08-10
> **Response to Reviewer Ms3t**
>
> Thank you for the detailed and thoughtful review. Below are responses to individual comments and questions.
>
> - **“I don't quite buy that 'using Z3' is the same as what Kahneman means by 'System 2’.”** We agree that dual process theory is much richer than what we have implemented here; our intention is not to claim that we have implemented a full model of dual process theory, rather that we are taking inspiration from dual process theory to address shortcomings of current neural systems in a relatively lightweight manner. We will revise our paper to make this distinction more explicit. Furthermore, while we don’t mean to claim that exact logical inference via an SMT solver is a precise model of human cognition, we note that humans are often able to perform logical reasoning, and we aim to use the symbolic tools at our disposal to build systems which imitate this behavior.
>
> - **Experimental validity:**
>   - **Human judgment experiment:** Your review argues that our human judgement experiment is not comparing the 'System 1 + 2' model with 'System 1'; but is instead comparing 'System 1 + 2' with 'System 1 - 2'. This comment brings up a good point: it is important to distinguish between the logic of the human behavioral study and how the magnitude of the effect is quantified. For the human behavioral study, we did not ask for human judgements in cases where the single-system and dual-system models produce the same utterance---practically, this would have meant asking (and paying for) humans to perform a two-alternative forced choice trial on two identical stimuli. Our methodology is valid, as long as the effect size can be quantified. The true effect size depends upon the prevalence of disagreements. These numbers are reported in Table 1. Here, we find the prevalence of disagreements is at least 14% - 57%, depending on the data regime. Thus, the model predictions do differ often, and we think these results are important and relevant for the community, as our approach has the promise of delivering on large scale improvements. We can clarify this reasoning in our revised paper, and we thank the reviewer for pointing this out.
>   - **gSCAN:** For the gSCAN experiment in Section 5, the LSTM-based System 1 component is actually critical for the performance of the model, and our System 2 model cannot solve the task on its own. The gSCAN domain requires a model to not only navigate to a target location specified by an instruction, but also to navigate in the specified manner, and perform the specified action. For example, given a particular gridworld state, the instructions “Push the small red circle cautiously,” “Walk to the small red circle while spinning”, and “Pull the small red circle while zigzagging” may refer to the same target location, but correspond to entirely different command-word sequences. Our System 2 model only tracks the agent’s target location, and has no knowledge of the manner and action specified, which are learned by the LSTM-based System 1 model. If, as suggested, the System 2 model was used to symbolically induce a command-word sequence, it would only induce a simple path from initial location to target location; specifically, it would fail on all tasks which specify a manner or action to be performed. We realize that our draft did not include a complete description of the gSCAN domain, for which we apologize. While space is limited, our revised paper will make these points clear. We also note that our dual-system model achieves performance gains over the neural baseline, even though the System 2 model encodes only some aspects of the full problem. We view this as a strength of our approach; it can be applicable in domains where a full symbolic treatment is not feasible.
>
> - **Citations:** We will change the citations to non-arxiv versions of the published papers, including for work cited in Kahenman’s 2011 book.
> - **Related work:** We view this work as an extension of generate and rerank, where the reranking is done via explicit logical constraints. From lines 99-102: “Reranking is generally used to improve outputs with respect to relatively broad, holistic criteria. Here, our goal is to make generation robust to particular types of logical errors by pruning with respect to explicit symbolic constraints.” We will also incorporate the related works you mentioned, thank you for those references.

---

### Official Review · Reviewer_jmVT · 2021-07-17

**Rating:** 5
**Confidence:** 4

**Summary:**

Inspired by the philosophical literature, this paper proposes to view human reasoning as consisting of two Systems: System 1 represents the "reflexive", everyday part of our reasoning, which makes sure we can do most of our tasks without having to deliberate about them. System 2 represents out "deliberative" part, which we activate when something surprising happens and we have to react in a different way than we are used to, and often involves some sort of logical reasoning. This view is popularized by Kahneman in his book "Thinking, fast and slow", and it provides an appealing way of separating the two main schools of thought in AI: the classical symbolic approach (System 2), and the currently very popular ML approach (System 1).

The authors of this paper observe that pre-trained models such as GPT-3 appear to lack System 2 abilities, and they propose an approach where they build a "minimal world model" for a specific domain using a logic. They use GPT-3 to rewrite problems to this logical language, and use a solver to verify whether the output from the pre-trained model is consistent. If not, the output is rejected and the next best one is tried.

They evaluate their approach on text question answering and generation (bAbI) and instruction generation (gScan), and show that their approach can increase the coherence and accuracy.

**Ethical Concerns:**

No ethical concerns

**Limitations And Societal Impact:**

I don't see problems here

**Main Review:**

The paper is clearly written, and the proposed approach is explained well. I could understand all details and the experiments make sense to me. I also think the topic is of high interest to the AI community. It is clear that pre-trained models have limitations, and the current paper addresses an important one, namely that they often make mistakes that we humans wouldn't make.

My main issue with this paper is that I am not convinced about the significance of the results and the usability of the framework in more general settings. I see the following problems:
1) Inference speed. Since the consistency check is not differentiable, it seems that one is required to run this at each inference step. When the complexity of the domain grows, computing whether an output is consistent according to the underlying logic can become a bottleneck, and I would have liked to see some more details on this.
2) Developing the logic. It seems to me that this approach will only work if the minimal world model is specified correctly. This works for the simple toy domains discussed in this paper, but for any realistic domain, developing a consistent logic is a major undertaking, and I feel the current approach sidesteps this problem by taking very simple domains. I therefore don't see how this approach can be useful in a more general setting.
3) Empirical results. I feel that the main reason the authors managed to get good empirical results is that they developed domain-specific logics and it is quite simple to rewrite sentence from the dataset to this logic (I guess one could almost handcode them). Therefore I wasn't particularly impressed by the results.

**Time Spent Reviewing:**

4

---

> ### Author Response · Authors · 2021-08-10
> **Response to jmVT**
>
> We would like to thank you for the thoughtful review.
> - **Inference speed:** The question on inference speed raises an important point. While inference speed may be a bottleneck, we imagine that these models will be most likely applied to logical inference problems for which longer compute times are necessary to ensure correctness. One weakness of current neural sequence generation systems (e.g., transformer-based language models) is that they cannot address situations which *require* more computation before outputting an appropriate response.
> - **World model design:** Our approach does require designing the world model for each domain. However, we do note that, for our approach to succeed, it does not necessarily require tracking all of the important state information; in the gSCAN domain, our dual-system model increases performance over the neural baseline, even though the System 2 model only encodes the goal location, which is just one aspect of the information needed to solve the problem (for more details, see the explanation in response to Reviewer Ms3t). Again, we view this as a strength of our approach; it can be applicable in domains where a full symbolic treatment is not feasible. We also believe this work serves as a proof of principle; as discussed in the limitations section, integrating this work with approaches for learning symbolic models could be a promising area for future research (Ellis et al. 2021, Michael and Valiant, 2008, and others).
> - **Empirical results:** We note that the improved performance on our domains is not a guarantee. For example, in the CLUTRR story generation domain, there are several aspects which must succeed in order for improved performance to be possible. First, the semantic parser needs to effectively parse constraints from the neural language model generations, and there is no guarantee it would do so accurately. Second, it’s not obvious that the system 1 neural generator can recover from incorrect generations; for this to be true, a non-violating utterance must have relatively high sampling probability under the model. Finally, even if the model’s filtering process is sound, it’s not clear that people would be sensitive to the changes it makes in generation, and that it would ultimately lead to more human-like generations.
>
> **References:**
> Ellis, K., Wong, C., Nye, M., Sablé-Meyer, M., Morales, L., Hewitt, L., ... & Tenenbaum, J. B. (2021). DreamCoder: bootstrapping inductive program synthesis with wake-sleep library learning. In Proceedings of the 42nd ACM SIGPLAN International Conference on Programming Language Design and Implementation (pp. 835-850).
>
> Michael, L., & Valiant, L. G. (2008). A First Experimental Demonstration of Massive Knowledge Infusion. In KR (pp. 378-389).

---

> > ### Comment · Reviewer_jmVT · 2021-09-13
> > **Response from reviewer**
> >
> > I appreciate the reviewers took the time to reply to all my comments. I agree the improved performance is not a guarantee, but still it does not convince me enough to change my rating.

---

### Author Response · Authors · 2021-08-10
**Overall response for Paper10040**

We very much appreciate the thoughtful comments from all of the reviewers.

We see that the reviewers’ comments mostly concern the novelty, significance, and applicability of the work, as opposed to its technical correctness.

There were some questions about technical correctness, which we believe we’ve addressed fully (in particular, see the response to Reviewer Ms3t, which discusses why the experimental methodology of the human judgement experiment is valid, and describes how the gSCAN experiment is also valid, because the gSCAN domain cannot be solved by the System 2 model alone).

In response to questions about novelty, significance, and applicability, there is inherent subjectivity to these judgements. We see our approach as simple, lightweight, and easy to implement using off-the-shelf tools. Despite this simplicity, our experimental results show improved coherent language generation (CLUTRR) and performance gains on an instruction-following benchmark (gSCAN). We do not think our approach is applicable to all sequence generation tasks, but hope it can be useful for domains where learning-based generation is necessary, but explicit logical constraints are also readily available.

We acknowledge that our approach is best suited for domains where a human can readily express relevant domain knowledge in symbolic form, effectively requiring a human in the loop. However, we point out that, in practice, transformer-based language generation systems such as GPT-3 are also used as augmentative tools with a human in the loop; it would generally be unwise to leave GPT-3 unattended or unsupervised in a mission-critical application. Our approach will minimize the amount of human intervention in some applications, reducing the need to monitor or filter individual responses.

---

### Decision · Program_Chairs · 2021-09-28

**Decision:**

Accept (Poster)

**Comment:**

This paper borrows insights from neuroscience and psychology, exploring how symbolic approaches can be combined with a large-scale neural language models to improve responses because systems like GPT3 make mistakes that humans would not. This work can be viewed in several different lights: evidence that the system1/system2 insight is useful for ML systems and should be explored more; further evidence that hybrid ML and symbolic approaches are promising; a way to improve (increasing coherence and accuracy) black-box ML systems for deployment. Indeed the paper speaks to many of these views, but the reviewers felt that the paper does none of them convincingly.

The reviewers flagged several issues including: speed of the inference step, that this is not really a compelling instantiation & evaluation of the sys1/sys2 concept, and the consistency in the parsing process. One reviewer was concerned that the work builds on and uses a closed source entry (GPT3) for both practical and ethical reasons. Several reviewers flagged useful related works whose discussion will strengthen the work. In the end all four reviewers---all working in related areas---all agreed on rejection. The main concern being scalability and generality of the proposed approach due to the need to hand-design design world models for each domain. The reviewers found the lack of general principles & advice offered for this critical step unconvincing. The experiments presented did not do enough to convince on this critical question. Further, the discussion revealed the reviewers were not satisfied with this as a first step---that too much was being left to future work---and that non-trivial revisions are needed. Besides these concerns the work is polished and he author response comprehensive. On balance, this work was close but just below the bar.



**Consistency Experiment:**

NeurIPS has a long history of experimentation. In 2014, NeurIPS ran an experiment in which 10% of submissions were reviewed by two independent committees to quantify the randomness in the review process. This year, we repeated a variant of this experiment to see how the quality of the review process has changed over time.  This paper was part of the experiment and was therefore assigned to two committees (consisting of reviewers, an Area Chair, and a Senior Area Chair) that reached independent decisions.  If both committees made the same recommendation, this recommendation was followed. If a single committee recommended acceptance, the paper was accepted (with the exception of a few cases in which the other committee identified what we considered a fatal flaw, e.g., an error in a key result).

This copy’s committee reached the following decision: **Reject**

The other committee assigned to the paper recommended **Accept (Spotlight)**.  You can find the other set of reviews, along with any follow up discussion with the authors here:
https://openreview.net/forum?id=uyKk_avJ-p4